# Meniscal tear outcome Study (METRO Study): a study protocol for a multicentre prospective cohort study exploring the factors which affect outcomes in patients with a meniscal tear

Imran Ahmed ,[1] Mike Bowes,[2] Charles E Hutchinson,[3] Nicholas Parsons,[3] Sophie Staniszewska,[3] Andrew James Price ,[4] Andrew Metcalfe [1,5]

For numbered affiliations see end of article.

**Correspondence to**
Imran Ahmed;
imran.ahmed4@nhs.net

## ABSTRACT

**Introduction** This study is designed to explore the baseline characteristics of patients under 55 years of age with a meniscal tear, and to describe the relationship between the baseline characteristics and patient-reported outcome measures (PROMs) over 12 months. Research has highlighted the need for a trial to investigate the effectiveness of arthroscopic meniscectomy in younger patients. Before this trial, we need to understand the patient population in greater detail.

**Methods and analysis** This is a multicentre prospective cohort study. Participants aged between 18 and 55 years with an MRI confirmed meniscal tear are eligible for inclusion. Baseline characteristics including age, body mass index, gender, PROMs duration of symptoms and MRI will be collected. The primary outcome measure is the Western Ontario Meniscal Evaluation Tool at 12 months. Secondary outcome measures will include PROMs such as EQ5D, Knee Injury and Osteoarthritis Outcome Score and patient global impression of change score at 3, 6 and 12 months.

**Ethics and dissemination** The study obtained approval from the National Research Ethics Committee West Midlands—Black Country research ethics committee (19/WM/0079) on 12 April 2019. The study is sponsored by the University of Warwick. The results will be disseminated via peer-reviewed publication.

**Trial registration number** UHCW R&D Reference: IA428119. University of Warwick Sponsor ID: SC.08/18–19

## Strengths and limitations of this study

► Prospective study design: The study is prospectively designed which will reduce the risk of recall bias among participants.
► Multicentre design: By recruiting participants from multiple centres across the country, we are obtaining data from a range of physiotherapists and surgeons. This will ensure that the findings are representative of current UK National Health Service practice.
► Clearly defined outcome measures: We have pre-selected our patient-reported outcome measures (PROMs) which are widely used in previous meniscal research. This will enable comparisons with previous studies and also pooling of results in future reviews.
► Comprehensive clinical and radiological assessment of participants: We are including several important baseline features. This includes clinical history directly related to current British Association for Surgery of the Knee treatment guidelines. We are also the first study to explore the relationship between WORMS score and PROMs in patients with a meniscal tear.
► Absence of objective clinical outcome measures: A limitation of this study is that we are not reporting clinical outcome measures, for example, range of motion in the knee, muscle strength and return to sport.

## INTRODUCTION

Meniscal tears affect 60–70 per 100 000 of the population in the UK and account for 70 000 UK hospital admissions per year.[1 2] Over recent years, there has been a substantial increase in meniscus-related literature, with a number studies questioning the effectiveness of surgical management.[3–5]

The meniscus is a highly specialised c-shaped structure located within the knee.[6] It has an important role in distributing load across the knee joint, preventing damage to the articular cartilage.[7] The meniscus is susceptible to tears or damage. Meniscal tears typically result from high energy twisting injuries, such injuries often occur during sporting activities.[8 9] Meniscal pathology is also common in older people and is closely associated with osteoarthritis (OA) and

age-related degeneration, which can lead to changes in the histology of the menisci and their mechanical properties.[10]

The current management of meniscal tears includes non-surgical options such as provision of advice and observation, physiotherapy or pharmacological options such as anti-inflammatory medication or intra-articular corticosteroid injections. Surgical options include meniscal repair (in limited cases) or arthroscopic partial meniscectomy (APM), a keyhole procedure performed with the intention of removing the torn or unstable meniscus which may be thought to cause pain or mechanical symptoms, for example, locking.[11]

Previous randomised controlled trials (RCTs) have found no evidence for a difference in outcome between APM and physiotherapy in patients with a meniscal tear and coexisting arthritis.[12 13] This led to a change in National Institute for Health and Care Excellence guidelines advising against arthroscopic surgery to treat meniscal tears in the context of established arthritis.[14] Abram *et al* performed a systematic review including studies comparing arthroscopic meniscectomy versus a comparator in patients with a meniscal tear.[4] The authors included 10 studies for analysis and performed subgroup analysis for patients without OA. The authors reported a small improvement in knee pain for all patients with a meniscal tear in the meniscectomy group compared with the non-surgical group (standardised mean difference (SMD) 0.22 95% CI 0.04 to 0.4). Three studies reported pain scores in patients without OA, in which the authors found an increased improvement in pain scores in the meniscectomy group (SMD 0.35, 95% CI 0.04 to 0.66). There was a similar improvement in knee function in the meniscectomy group in patients without OA compared with all patients undergoing meniscectomy (SMD 0.18 vs SMD 0.3).

Despite the evidence challenging the clinical effectiveness of APM, there has been a 22% increase in the number of APMs performed over the last 20 years, from 151/100 000 in 1997 to 184/100 000 in 2017.[15–17] Previous literature has provided evidence on the lack of effectiveness of APM in patients with a degenerative knee or arthritis as the symptoms may be caused by the degeneration and not the meniscal tear.[18 19] UK national consensus statement from British Association for Surgery of the Knee (BASK) has highlighted the importance of patient factors which are important in the management of meniscal tears. The guidelines state patients with a particular pattern of meniscal tear visible on MRI ('a meniscal target') and the corresponding pattern of symptoms or signs should be recommended for non-urgent arthroscopy. Furthermore, patients should undergo a period of non-operative management before referral for APM.[20] These guidelines also suggest that radiology investigations should be included in treatment decisions, with certain patterns of tears being 'target' lesions.[20] A future cohort study will need to assess the importance of the MRI tear pattern as an outcome predictor.

A well-designed cohort study needs to be performed with patients being recruited at the point of diagnosis or presentation to an orthopaedic specialist. This will allow researchers to include patients being managed non-operatively. By including MRI analysis in a potential predictive model, this will produce a comprehensive model to identify which features can predict variability in patient-reported outcome measures (PROMs).

## Study aim
To describe the baseline characteristics and imaging findings of young patients with a meniscal tear and to explore the relationship, if any, between these baseline features and 12-month outcomes.

We will address the following research objectives:
1. Describe the baseline characteristics, pattern of symptoms and imaging findings of younger people (aged <55 years) with a meniscal tear (study population).
2. Explore the relationship between these features and 12-month PROMS.

## METHODS AND ANALYSIS
A multicentre prospective cohort study will be performed. The study obtained approval from the National Research Ethics Committee (NRES) West Midlands—Black Country research ethics committee (19/WM/0079) on 12 April 2019. All patients presenting to a secondary care centre (elective knee clinic or acute knee clinic) with a MRI confirmed meniscal tear, under the care of orthopaedic consultants at any of the participating sites, will be invited to take part in the study. The following eligibility will be implemented for patient selection.

Inclusion criteria:
► Age between 18 and 55 years.
► Presence of an MRI confirmed meniscal tear.
► Provision of informed written consent.

Exclusion criteria:
► Anterior cruciate ligament or other major knee ligament injury. This does not include a previous unrelated healed medial collateral ligament tear or a meniscal root tear (which is considered a type of meniscal tear in this study).
► Associated intra-articular fracture of the tibial plateau or femur. Previous fractures not thought to be related to the tear are not an exclusion criteria for the study.
► Previous knee surgery, for example, unicondylar knee replacement; total knee replacement; knee arthroscopy; meniscal repair or meniscectomy.
► Previous entry into the present study (ie, the other knee).
► Participant is unable to undertake study procedures.

## Participant identification and screening
Potential participants will be identified by a member of the attending clinical team or a member of the clinical research team by screening clinic lists in secondary or intermediate care clinics. This will involve a member of

the clinical team reviewing letters for the upcoming knee or orthopaedic clinics to identify potential participants.

Eligibility will be confirmed based on assessment and standard care MRI. In normal clinical practice, MRI is a requirement for diagnosis of a meniscal tear, therefore, all participants entering the study will already have had one at the time of entry into the study.

All individuals who meet the study entry criteria will be checked for eligibility and recorded on the monthly screening log. Eligible potential participants who are willing to be approached by a suitably trained member of the research team will be provided with verbal and written information about the study, and will have the opportunity to discuss and ask questions.

## Baseline data collection

Once participants have provided informed consent, baseline data will be collected by the research team. Baseline characteristics such as age, gender, body mass index and date of potential injury and duration of symptoms will be collected. Baseline PROMs will be collected at recruitment in order to assess changes in PROMs at 12 months. Clinical features which are consistent with a treatable meniscal lesion, outlined by the BASK consensus meeting, will also be collected from the attending clinician (for a locked knee) and the participant (for locking and catching).[20] These include the presence of:

► Locked knee: sudden onset, complete mechanical block to flexion or extension of the knee, detected on clinical examination and which does not resolve despite adequate analgesia.
► Locking: an intermittent block to normal range of movement of the knee (commonly a block to extension) with an associated unlocking movement. Knee returns to near normal after unlocking.
► Catching: the sensation of something intermittently out of place in the knee and interfering with joint movement.

These will be assessed on initial baseline questionnaires which will be provided by the host site. If the clinician documents specific examination findings (in particular alignment), this will be collected by the study team.

MRI scans will be analysed to assess:
1. Anatomical location and tear pattern. Tear patterns include the following which have been recognised by BASK as potentially treatable 'meniscal targets'.[20]
   a. Bucket handle tear: a longitudinal tear involving more than 25% of the meniscus length (can be displaced or undisplaced).
   b. Displaced meniscal tear: fragments are displaced from their usual anatomical position.
   c. Meniscal root tear.
   d. Radial tear: a vertical tear which may or may not extend into the meniscocapsular junction.
   e. Horizontal tear.
   f. Complex tear: a meniscal lesion with more than one place of tear in continuity.

2. Whole-Organ MRI Score (WORMS): This is a semiquantitative, multifeature scoring method for wholeorgan evaluation of the knee using conventional MRI. It is based on 14 features include cartilage and meniscal integrity.[21]
3. Bone area: analysis of bone area and cartilage volume using proprietary semi-automated segmentation software in collaboration with IMorphics (IMorphics, Stryker House, Hambridge Road, Newbury, Berkshire). Previous research has demonstrated that change in bone area is a more sensitive marker of OA than cartilage thickness or joint space narrowing. This change in bone area could lead to the development of meniscal tears as flattening of the femoral condyles could potentially reduce the space available for the meniscus increasing the likelihood of meniscal damage.[22]

Each MRI will be reported by an orthopaedic surgical trainee (IA) trained to report WORMS scores and a consultant radiologist (CEH).

## Primary outcome

Western Ontario Meniscal Evaluation Tool (WOMET) at 12 months. This is a meniscal tear disease-specific quality of life measure developed in collaboration with patients in 2007. It consists of 16 questions (items) focusing on three domains: (1) physical symptoms, (2) sport/recreation/work/lifestyle and (3) emotions. Items are assessed on a Visual Analogue Scale (VAS), summed and reported as a percentage of the maximum score of 1600.[23] A recent systematic review of PROMs used for meniscal research demonstrated that WOMET had the strongest evidence for content validity.[24] It is also one of the only PROMs developed with the involvement of patients with meniscal tears as their primary diagnosis.

## Secondary outcomes

► WOMET: collected at baseline, 3 and 6 months: as outlined above.
► Knee injury and Osteoarthritis outcome score 4 (KOOS)[25]: The KOOS questionnaire was developed in 1998 as a knee injury-specific outcome measure of patients at risk of developing arthritis.[26] Domains are pain, symptoms, functional status, sports activity and quality of life. It is used extensively in knee injury research, and has been the primary outcome in the majority of clinical trials studying the treatment of meniscal tears. In previous meniscal tear research a shorter form of KOOS ($KOOS_4$) has been widely used.[25] This focuses on four out of five of the domains pain, symptoms, sport and recreational function and quality of life. Standardised answer options are provided using a Likert scale and each question is assigned a score of 0–4. A score of 100 indicates no symptoms whereas a score of 0 indicates extreme symptoms. This outcome tool can be used as a postal or electronic survey and both versions are comparable with regard to psychometrics.[27] KOOS4 will be collected at baseline and 12 months only.

► EuroQol (EQ-5D-5L (EuroQol-5 domains- 5levels): This is a validated measure of health-related quality of life, consisting of a five-dimension health status classification system and a separate VAS. EQ-5D is applicable to a wide range of health conditions and treatments and provides a simple descriptive profile and a single index value for health status, range from −0.594 to 1, and anchored at 0 (death).[28 29] EQ-5D is primarily designed for self-completion by respondents and is ideally suited for use in postal surveys, in clinics and face-to-face interviews.[30 31] It is cognitively simple, taking only a few minutes to complete.[29 32] This will be collected at baseline, 3 months, 6 months and 12 months.

► Patient global assessment of change (PGIC): A simple 7-point scale assessing participant perception of improvement.[33] A standardised question will be used to explore the change in activity limitations, symptoms and quality of life in the painful knee. Responses will range from 'no change or condition has worsened' to 'a great deal better. This will provide valuable information for a future large scale trial by providing information on the minimally clinical important difference in WOMET score for this patient population. This will be collected at 3 months, 6 months and 12 months.

► Surgery: We will also collect data on whether participants undergo surgery for their meniscal tear. At each time point (3, 6 and 12 months) patients will be asked if they underwent surgery. We will continue to follow up patients after surgery to determine their outcome in a pragmatic fashion, as would happen in the conservative arm of a large trial.

These PROMs will be collected at baseline, 3, 6 and 12 months from the date of recruitment. Data will be collected using questionnaires designed with patient and public involvement. This data will be collected by a number of methods (either postal, telephone, electronically or face to face) dependent on participant preference. If a participant requests face-to-face consultation or has difficulty reading or writing, a clinic visit will be arranged.

## Sample size calculation

Previous experience[34] for studies of this type, where statistical models have been built to explain variation in PROMs, suggests that they rarely explain more than 10%–20% of the variation in outcomes ($R^2$) at 12 months. Using a large (and therefore rather conservative) estimate for the numerator number of df of 25 (ie, the model complexity) for a putative F test of the model significance and an $R^2$=0.1, suggests a sample size of 160 for 90% power at the 5% significance level (https://cran.r-project.org/web/packages/pwr/; pwr.f2.test).[35] Experiences of previous such studies in similar settings have reported follow-up rates of 80%. Assuming lost to follow-up of 20%, 200 participants will be required for this study.

## Statistical analysis

Summary statistics will be used to describe baseline data. We will report means and SDs (95% CIs) for approximately normally distributed data and for other cases report the median and IQR. Mean 12 months PROMs scores will be reported for the whole cohort. We will also report mean 12 months PROMs for both the non-operative and operative group. Pearson correlation coefficients will be used to assess evidence for associations between baseline features and 12 months PROMs scores. A model will be developed relating the response variable (12 months PROM) to a number of explanatory variables (eg, data collected at baseline). The type and structure of the models used will depend on the distribution of the response variable, and the nature of the association between the response and explanatory variable. A linear model will be the initial choice in order to predict the response variable (12 months PROM) as a linear weighted sum of the explanatory variables. The 'weights' or parameters of this model are estimated using regression analysis; providing us with regression coefficients (parameter estimates), that characterise the association.

A sensitivity and specificity-based longitudinal anchor method will be used to calculate the minimum clinically important differene (MCID). The change in WOMET scores will be calculated for each group on the PGIC scale. This will be done by subtracting the 12-month WOMET from the baseline score. A mean change in WOMET score will be associated with each anchor on the PGIC as follows:

1. No change (or condition has got worse).
2. Almost the same, hardly any change at all.
3. A little better, but no noticeable change.
4. Somewhat better, but the change has not made any real difference.
5. Moderately better and slight but noticeable change.
6. Better and a definite improvement that has made a real and worthwhile difference.
7. A great deal better and a considerable improvement that has made all the difference.

We will then use the receiver operator characteristics (ROC) to determine the MCID with equal sensitivity and specificity to discriminate between the changed group[5–7] and the unchanged group.[1–4] The area under the ROC curve represents the probability the score discriminates between the unchanged and the changed group. A probability of >0.7 was deemed acceptable.

This methodology has been previously used in orthopaedic research and has been described in previous literature.[36 37]

Model fitting and analysis will be undertaken in R (R Core team (2013) R Foundation for statistical computing, Vienna, Austria).[38]

## DISCUSSION

This prospective cohort study aims to describe the baseline characteristics of young people (age <55) with a

meniscal tear and to explore the relationship between these features at outcome. This study has a role in informing treatment decisions by identifying in certain patient features can predict treatment outcome. There is a view among researchers and clinicians that surgery may be beneficial among a particular subset of patients and further research is needed to identify this subset.[4] This work will assist in the future planning of an RCT by further exploring the study population. The study aims to identify the patients that improve with both operative and non-operative management and whether there are certain factors which influence the response to either treatment. It may be that in a population of patients who do not improve with non-operative care and do not have features of arthritis, that a study is required to assess the effectiveness of surgery, but before this can be performed, we need a much better understanding of the characteristics of this population and how they might be defined. This cohort study will also inform a future trial by providing an insight into follow-up rates in this younger population, efficient means of collecting outcome data, and also providing an understanding of the MCID of the WOMET score.

This study aims to build on a previous cohort study which was performed in order to identify features which may predict variability in PROMs for patients following arthroscopy. The authors included 18 preoperative factors and found the strongest predictive factors were no previous meniscal surgery and more severe preoperative knee symptoms. However, factors such as tear pattern or structural OA were not assessed as preoperative imaging was not available to the authors, and so our study will answer a related, but different question. The models overall predictive performance was very low (optimum adjusted $R^2=0.080$) suggesting the included factors had little relevance to outcome.[5] The authors only included patients undergoing operative intervention, work is needed to identify outcomes in all patients that are managed with a meniscal tear, in order to identify which patients performed well with non-operative management and which perform poorly.

An upper age limit of 55 was selected, as the mean age in a recent meta-analysis of trials in meniscal tears. As the one of the purposes of the study is to explore the relationship between baseline data and outcome, we will explore the effects on age. Age was not an important factor in the recently published analysis of the Knee Arthoscopy Cohort Southern Denmark (KACS) cohort and while it may be relevant, it may also be a poor proxy for other relevant factors, such as underlying OA or meniscal degeneration. This will be explored in this study and, if age is important, the study will allow us to set evidence-based thresholds for future research.

Strengths of the METRO cohort include its prospective nature, minimising the risk of participant recall bias. We are also reporting outcomes in all patients with a meniscal tear including both the operative and the non-operative group, this ensures the study results are more representative of all patients with a meniscal tear. The multicentre national nature of this study ensures that the data included are generalisable to current UK practice, especially after publication of the national treatment guidelines.[20] Further strengths include the use of MRI based assessment of tear pattern, arthritis and bone shape change which could be an important predictive factor for outcome.

Limitation of this study design in comparison to previous research is our sample size is smaller than a previous study which did not identify any predictive factors,[5] however, this study did not include MRI data and we have included a thorough sample size calculation with a conservative estimate of the number of df. In addition, this observational study will provide an insight into associations between patient factors and PROMs, however, as the study is observational the findings will be utilised for generation of new hypotheses and causal inference will be limited, and will require further study in RCTs. A multicentre RCT is currently recruitment comparing surgery versus physiotherapy, however, the eligibility criteria do not take into account presence of OA on MRI.[39] This cohort study will explore whether that is an important factor in patient outcome.

There is a need for further RCTs in a specific subset of patients where surgery may be of some benefit. However, we believe this study is in important, not only in informing current treatment pathways and adding further evidence to national guidelines but also contributing considerably to the planning of a future randomised trial assessing the effectiveness of arthroscopic meniscectomy in this young population.

## PATIENT AND PUBLIC INVOLVEMENT

Eighteen patients with knee pain or OA were consulted prior to study set up. Twelve of these patients had a meniscal tear. The patient and public involvement (PPI) group provided valuable input on study setup such as sources of recruitment, completion of case report forms, completion of questionnaires and how to conduct follow-up. These patients also reviewed the language used in patient information sheets and consent forms to ensure it was appropriate for the target population.

We are in the process of setting up a PPI reference group of five patients with a meniscal tear. They will help address any issues which arise during the study.

Patient and public members will be involved in the interpretation of the results and facilitate the dissemination of the study results by helping to prepare patient leaflets, manuscripts for peer-reviewed publication and dissemination through social media.

Patient and public members will also be regularly updated on study progress and will provide advice on issues including recruitment.

PPI will be reported using Guidance for Reporting Involvment of Patients and the Public 2 (GRIPP2) reporting checklist.[40]

## ETHICS AND DISSEMINATION

The study obtained approval from the National Research Ethics Committee (NRES) West Midlands—Black Country research ethics committee (19/WM/0079) on 12 April 2019.

We will aim to publish the results in a peer-reviewed journal, within 12 months of the study completing (recruitment began in April 2019 and will continue to November 2020). The results of the study will also be disseminated via patient information material produced in collaboration with the PPI group. All key study findings will be presented at national and international conferences, for example, British Orthopaedic Association or BASK, and the American Academy of Orthopaedic Surgeons (AAOS).

**Author affiliations**
¹Warwick Clinical Trials Unit, University of Warwick Warwick Medical School, Coventry, UK
²IMorphics Limited, Manchester, UK
³Warwick Medical School, University of Warwick, Coventry, UK
⁴Nuffield Department of Orthopaedics, Rheumatology and Musculoskeletal Sciences, University of Oxford, Oxford, UK
⁵Trauma and Orthopaedics, University Hospitals Coventry and Warwickshire NHS Trust, Coventry, UK

**Acknowledgements** We would like to acknowledge the support of: 1. The University of Warwick (Lead Sponsor). 2. University Hospital Coventry and Warwickshire (Lead NHS site). 3. Members of the PPI group. 4. Research and Clinical staff at all recruiting site: University Hospital Coventry and Warwickshire, George Elliot Hospital NHS Trust, University Hospitals Birmingham NHS Trust, Royal Orthoapedic Hospital, Birmingham Community Healthcare NHS trust, Robert Jones and Agnes Hunt NHS trust, Oxford University Hospitals, Imperial College NHS trust and North Bristol NHS trust.

**Contributors** IA: study conception, study design, drafted and reviewed final manuscript. MB: study design, drafted and approved final manuscript. CEH: study design, drafted and approved final manuscript. NP: study design, drafted and approved final manuscript. SS: Study design, drafted and approved final manuscript. AJP: study design, drafted and approved final manuscript. AM: chief Investigator, study design, drafted and approved final manuscript.

**Funding** The study protocol represents research funded by a National Institute for Health Research (NIHR) Doctoral Fellowship Award (DRF-2018–11-ST2-030). The study is sponsored by the University of Warwick. The study sponsor provides ultimate approval of all new versions of the protocol before they become live. Both the funder and sponsor are required to provide final approval before publication of any study material.

**Disclaimer** The study funder and sponsor had no role in the study design; the collection, analysis, or interpretation of data; the writing of the report; or the decision to submit for publication. The researchers are independent and the views expressed are those of the authors and not necessarily those of the NHS, the NIHR or the Department of Health.

**Competing interests** MB is a senior director of clinical applications at IMorphics limited.

**Patient and public involvement** Patients and/or the public were involved in the design, or conduct, or reporting, or dissemination plans of this research. Refer to the Methods section for further details.

**Patient consent for publication** Not required.

**Provenance and peer review** Not commissioned; externally peer reviewed.

**ORCID iDs**
Imran Ahmed http://orcid.org/0000-0003-2774-9954
Andrew James Price http://orcid.org/0000-0002-4258-5866
Andrew Metcalfe http://orcid.org/0000-0002-4515-8202

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
