## [Reviewer comments · BMJ Open]

ARTICLE DETAILS

TITLE (PROVISIONAL)	The Meniscal tear outcome study (METRO Study): A study protocol for a multicentre prospective cohort study exploring the factors which affect outcomes in patients with a meniscal tear.
AUTHORS	Ahmed, Imran; Bowes, Mike; Hutchinson, Charles; Parsons, Nicholas; Staniszewska, Sophie; Price, Andrew; Metcalfe, Andrew

VERSION 1 – REVIEW

REVIEWER	Jonas Bloch Thorlund University of Southern Denmark
REVIEW RETURNED	08-Apr-2020

GENERAL COMMENTS	Thank you for the opportunity to review this protocol paper. I think it is very nicely outlined what the investigators are aiming at and the rationale. Also, I think this cohort will contribute with new and additional data to the already existing knowledge. Especially, the real challenge of identifying specific sub groups of patients, which so far has not been as successful as one would image. I understand that the study is ongoing, and since the study and rationale is really nicely outlined I only have very few comments for the authors to consider. Introduction: Lines 5-6: 70,000 hospital admission, where? Please provide context. The introduction is very thorough and detailed, but also quite long. I think it would be possible for the authors to simplify and shorten to improve readability without losing the details for the rationale of the study. Methods: I think it is very nice that the authors try to differentiate between 'locked knee', 'locking' and 'catching' – these are things that are commonly confused. I think the challenge will be for the patient to understand the difference, but I really like the idea. MRI: I wonder to what extent it is actually possible to reliably assess the tear pattern on MRI. Perhaps a study should be embedded comparing the 'tear' scoring on MRI with the actual findings among those undergoing surgery. Also, as I understand clinical MRI's will be used. I guess these may differ between between recruitment sites? What consequences will this have for the MRI scoring (WORMS) and assessment of tear patters?
---

	Responses to PROMS may be affected by the method of delivery and the setting in which these are filled out. Have the authors considered this when selecting a variety of delivery methods? Or do the authors have any knowledge about the potential impact? (clearly, I understand that this may be of pragmatic reasons) Despite the efforts of making a sample size estimation the intended number to be recruited (i.e. n=200) seems actually quite small if the aim is to predict outcome based on baseline data, especially given that previous attempts with a cohort 3 times the size was not very successful. I think this can turn out to be a major limitation of the study. (I understand the study has begun, but recruitment could potentially be prolonged)
--	---

REVIEWER	Rudolf Poolman and Victor van de Graaf LUMC Leiden and OLVG, Amsterdam
REVIEW RETURNED	22-Apr-2020

GENERAL COMMENTS	I would like to congratulate the team with this very interesting METRO project and the first author with his doctoral fellowship award. While the treatment of degenerative meniscal tears has gained much attention during last decade, very little is known with regard to the treatment of meniscal tears in younger patients indeed. Publication of research protocols is of the highest importance to improve transparency. Although, I am convinced of the relevance of this project, there are several issues that need to be addressed: The authors aim to publish their study protocol. However, I was unable to retrieve the original protocol through the registration details (I was only able to retrieve the following: https://www.fundingawards.nihr.ac.uk/award/DRF-2018-11-ST2-030). Please add the original trial protocol (with the document date) to the submitted manuscript which is required. Is the cohort registered in an online registry? I have some conflicting thoughts with regard to the set age limit of 55. What was the rationale of the authors to choose include patients up to 55 years of age? Since this research project aims to focus on a subgroup of (younger) patients, which is little known of, I was surprised to see the upper limit of 55. The published RCTs included patients from 45 years old (vdGraaf,Katz,Gauffin) and even from 35 years (Kise,Sihvonen). Furthermore, Rongen and colleagues demonstrated that the vast majority of procedures (76%) is performed in patients over 40 years of age, with the highest number of procedures performed in the age group of 50-55. In my opinion, the most relevant age group for this project, 20-40 years, represents only 20% of meniscal surgeries (Rongen JJ et al KSSTA 2018). Therefore, I have serious concerns of the authors will be able to answer their proposed research questions. Will you stratify for age to avoid an overrepresentation of patients from 45-55 years in your study group? Abstract - No comments, expect for the word count that is missing. Introduction:
--

- The authors mention that previous trials have demonstrated no evidence to justify the use of APM. However, they subsequently only refer to 1 study. Please change.
- The authors claim that the numbers of APM have increased over the past 20 years. However, they refer to 3 studies, the first study of their own group (in which they conclude: APM rates increased about 130% overall but have declined recently), and 2 other studies that included numbers only up to 2011. Please change.
- "Previous literature has provided evidence on the lack of effectiveness of APM in patients with a degenerative knee or arthritis as the symptoms may be caused by the degeneration and not the meniscal tear." Please add reference.
- International consensus from specialist knee societies has highlighted the importance of patient factors which are important in the management of meniscal tears. Please add reference, and if you choose to add the reference from your own groups (17), change 'international' to 'national'.
- "For example, patients with a particular pattern of meniscal tear visible on MRI ('a meniscal target') and the corresponding pattern of symptoms or signs may benefit from APM." – there is hardly any robust evidence for this. Please delete this statement.
- "Previous evidence supports the view that APM may not be beneficial in patients with meniscal tears and degeneration or arthritis within the knee joint (18)." This is incomplete. Several RCT have published their results and found no difference between APM and PT (or SHAM surgery) in patients with meniscal tears (Katz, Sihvonen, Yim, Kise, Herrlin, vdGraaf, Gauffin). None of the studies found clinically relevant differences between groups (with or without OA).
- There is a large number of self-references (11 times to 6 studies). For example, "This could explain why the literature reports that up to 30% of patients cross over into the surgery group (4)." Why do refer to you own systematic review here? This does not seem logic.
- "The Knee (BASK) guidelines suggest that radiology investigations should be included in treatment." please add reference.

Methods:

- How will the authors determine the level of osteoarthritis? Only on MRI (which is known to give an overestimation and also is highly unfeasible due to its extensive nature, costs, heterogeneous outcome reporting)? or will there also be an X-ray?
- Who will report the MRI? Several radiologists or only one or two that are trained according to the study protocol?
- "PROMs will be collected at recruitment, as a baseline for PROMS at 12 months (primary outcome). With follow up will be at three months, six months and 12 months." Please change sentence for better understanding.
- Can the authors explain why they chose for the KOOS instead of the IKDC as secondary outcome? If I interpret their own systematic review (and also the literature), I have come to conclude that the IKDC performs better on all domains compared to the KOOS. Most importantly, it is less than half of the length of the KOOS.
- Sample size calculation: why do the authors not account for crossovers in their sample size calculation?
- Please elaborate on how the MIC will be determined. Will ROC curves be determined for an optimal cut-off point? This needs more attention. The difference in the mean WOMET score will represent the MCID. I am really not convinced that this is the most valid method. Please check the COSMIN guidelines, as an incorrect

	calculation of the MIC will be of huge impact on the interpretation of the study results. Patient involvement: - Well done! My compliments that you have been able to involve patients from the start. - “Eighteen patients with knee pain or osteoarthritis” Did these patients have a meniscal tear? This information is imperative. Data management: - “Participants will be identified by ID number initials.” Who is responsible for the ID, what will the ID comprise of (any retrievable information?) And who will have access to the ID key? Please elaborate on this. Discussion section is missing. Please add this section and add relevant information, such as strengths/limitations, comparison with literature (there is one trial that is currently analyzing their results (the STARR trial), which has the same target group; please cite this study. Please stress that this observational study can generate hypothesis for future studies. Causal inference is limited “it will also allow a large randomised study to be planned to determine the best treatment for young people who are diagnosed with meniscal tears.” Why do the authors not elaborate on this? It seems only reasonable to at least mention this as aim of the study, the discussion section seems appropriate for this.”
--	---

VERSION 1 – AUTHOR RESPONSE

Reviewer: 1

Reviewer Name: Jonas Bloch Thorlund

Institution and Country: University of Southern Denmark

Please state any competing interests or state ‘None declared’: None to declare

Please leave your comments for the authors below

Thank you for the opportunity to review this protocol paper. I think it is very nicely outlined what the investigators are aiming at and the rationale. Also, I think this cohort will contribute with new and additional data to the already existing knowledge. Especially, the real challenge of identifying specific sub groups of patients, which so far has not been as successful as one would image. I understand that the study is ongoing, and since the study and rationale is really nicely outlined I only have very few comments for the authors to consider.

Thank you for your kind comments and also taking the time to review the manuscript.

Introduction:

Lines 5-6: 70,000 hospital admission, where? Please provide context.

This has now been made clear. The data relates to UK admissions and incidence.

The introduction is very thorough and detailed, but also quite long. I think it would be possible for the authors to simplify and shorten to improve readability without losing the details for the rationale of the study.

The introduction has been shortened to improve readability. We have now included a discussion section, allowing some of this information to be transferred to the discussion section.

Methods:

I think it is very nice that the authors try to differentiate between 'locked knee', 'locking' and 'catching' – these are things that are commonly confused. I think the challenge will be for the patient to understand the difference, but I really like the idea.

Thank you for this feedback, we tried to match our data collection sheet with the BASK management guidelines as much as possible. We recognise the challenge you mention, in our patient facing questionnaires we have taken particular effort to use language suitable for patients in order to avoid this confusion. We have trialled the questionnaire with our participants who felt it was clear and easy to interpret.

MRI:

I wonder to what extent it is actually possible to reliably assess the tear pattern on MRI. Perhaps a study should be embedded comparing the 'tear' scoring on MRI with the actual findings among those undergoing surgery.

We acknowledge that it may be difficult to reliably assess tear pattern. Our rationale for including this, other than to keep it pragmatic, is because the BASK guidelines recommend surgery in certain 'target lesions'. As clinical decisions are, by necessity, made on MRI findings, this is the focus of our research. If there is no relationship between MRI tear pattern and outcome, it should not be used regardless of its pathological accuracy. A consultant radiologist and an Orthopaedic registrar trained to interpret MRI knees will independently assess MRIs. We also take on board your advice on an embedded study and will explore whether it is feasible, given the multi-centre design, to do this using this cohort.

Also, as I understand clinical MRI's will be used. I guess these may differ between between recruitment sites? What consequences will this have for the MRI scoring (WORMS) and assessment of tear patterns?

We aimed to use clinical MRI scans as this is the type of imaging modality that will be used for future treatment decisions in clinical practice. To our knowledge it should have minimal impact on assessment of tear pattern as currently radiologists are using standard MRIs for this decision.

For WORMS scores: WORMS scores have previously been assessed in research studies using imaging taken with MRI using conventional pulse sequences and a clinical 1.5 T MRI system. This is similar to the sequence and systems used in standard NHS practice.

<https://pubmed.ncbi.nlm.nih.gov/14972335/>

Responses to PROMS may be affected by the method of delivery and the setting in which these are filled out. Have the authors considered this when selecting a variety of delivery methods? Or do the authors have any knowledge about the potential impact? (clearly, I understand that this may be of pragmatic reasons)

Pragmatic aspects and the need to ensure high rates of follow-up have determined this decision. The main reason for using different delivery methods is based on the age of the population. Previous studies have demonstrated lower follow up rates 20-30% when one method of follow up is used. As a result we wanted to use a combination of follow up methods to improve our follow up rate. With respect to KOOS, a previous study has shown that the subscale scores did not differ between postal and electronic.

<https://www.ncbi.nlm.nih.gov/pmc/articles/PMC4116261/>

Despite the efforts of making a sample size estimation the intended number to be recruited (i.e. n=200) seems actually quite small if the aim is to predict outcome based on baseline data, especially given that previous attempts with a cohort 3 times the size was not very successful. I think this can turn out to be a major limitation of the study. (I understand the study has begun, but recruitment could potentially be prolonged)

We acknowledge that the sample size is smaller than the KACS cohort although we are hoping to learn from the results of the KACS cohort which was a substantial achievement. The formal power analysis presented in the manuscript is motivated by our main aim of developing a model that is able to explain a very modest part of the variation (10%) in the reported outcome data (WOMET at 12 months). We believe that this will be achievable with our proposed sample size of 200 participants. The main determinant in successfully achieving this aim will be in large part determined by the quality of the available data and the strength of the association, rather than simply the numbers recruited into the study. The construction of useful models in this context is predominantly about quantifying the magnitude of the associations between the baseline data and future outcomes. We are in particular interested in MRI findings (tear pattern, WOMS, bone shape) which were not included in the KACS cohort and may have a particularly strong association with the outcome. If this is the case, then we would expect to be able to be able to develop models with practically useful predictive power with a relatively modest sample size. If the MRI data do not have an important effect on the outcome, this will itself be a clinically important finding. We also have to act within our funding and study management plan, but we will consider extending recruitment if this is felt to be needed and practically feasible.

Reviewer: 2

Reviewer Name: Rudolf Poolman and Victor van de Graaf

Institution and Country: LUMC Leiden and OLVG, Amsterdam

Please state any competing interests or state 'None declared': None

Please leave your comments for the authors below

I would like to congratulate the team with this very interesting METRO project and the first author with his doctoral fellowship award. While the treatment of degenerative meniscal tears has gained much attention during last decade, very little is known with regard to the treatment of meniscal tears in younger patients indeed. Publication of research protocols is of the highest importance to improve transparency. Although, I am convinced of the relevance of this project, there are several issues that need to be addressed:

The authors aim to publish their study protocol. However, I was unable to retrieve the original protocol through the registration details (I was only able to retrieve the following: <https://www.fundingawards.nihr.ac.uk/award/DRF-2018-11-ST2-030>). Please add the original trial protocol (with the document date) to the submitted manuscript which is required. Is the cohort registered in an online registry?

Thank you for your kind comments. We have attached the original protocol to the supplementary files. This is the protocol submitted to the ethics committee. We have also attached the fellowship application which is the original protocol submitted to funders. The cohort has not been registered to an online registry, as would normally be required for clinical trials, but in the interests of transparency we are publishing this detailed protocol paper before recruitment completes.

I have some conflicting thoughts with regard to the set age limit of 55. What was the rationale of the authors to choose include patients up to 55 years of age? Since this research project aims to focus on a subgroup of (younger) patients, which is little known of, I was surprised to see the upper limit of 55. The published RCTs included patients from 45 years old (vdGraaf,Katz,Gauffin) and even from 35 years (Kise,Sihvonen). Furthermore, Rongen and colleagues demonstrated that the vast majority of procedures (76%) is performed in patients over 40 years of age, with the highest number of procedures performed in the age group of 50-55. In my opinion, the most relevant age group for this project, 20-40 years, represents only 20% of meniscal surgeries (Rongen JJ et al KSSTA 2018). Therefore, I have serious concerns of the authors will be able to answer their proposed research questions. Will you stratify for age to avoid an overrepresentation of patients from 45-55 years in your study group?

Thank you for this comment. The purpose of this study is to examine the relationship between baseline factors and outcome, age will be considered amongst this. If age is important, it will be identified and we will know to use this as a way of focusing on a younger population, but we do not have robust cohort data to support different thresholds.

We believe the current literature focus on age is probably misplaced, as has been demonstrated by the recent, large KACS cohort. The recent study of Pihl et al (BJSM 2020, reference 5) found no relationship between outcome and multiple baseline factors including age. It is therefore likely that other pathological aspects such as the presence of associated osteoarthritis are more important and that age is being used as a poor proxy for the underlying condition of the knee or degeneration of the meniscus.

The rationale for choosing 55 and under was based on a previous systematic review (Abram et al 2019) where the mean age of included studies was 55. Therefore, the current body of clinical

trials do not provide a robust answer as at least half of the included patients were above the age of 55. We need to find an evidence based rationale for any age limits set for trials in the future and a study like this is well placed to do so.

This study will inform future research/ trials on patient population. One potential finding could be that the majority of patients over 40 have MRI features of degeneration which influences their outcomes. As a result they should not be included in a future trials assessing the clinical effectiveness of meniscal surgery.

In addition, we are not just exploring outcomes in surgical patients, we are interesting in describing outcomes of all patients being managed with a meniscal tear, including those managed non-operatively. We also aim to stratify for age by including age in our model to predict outcomes.

Whilst we appreciate the point the reviewer makes, we do not agree that it weakens the study (and, we should note, the cohort is already recruiting and we are unable to change the eligibility criteria).

Abstract

- No comments, expect for the word count that is missing.

Introduction:

- The authors mention that previous trials have demonstrated no evidence to justify the use of APM. However, they subsequently only refer to 1 study. Please change.

A further reference has been added.

- The authors claim that the numbers of APM have increased over the past 20 years. However, they refer to 3 studies, the first study of their own group (in which they conclude: APM rates increased about 130% overall but have declined recently), and 2 other studies that included numbers only up to 2011. Please change.

This has been clarified further. The number of APM increased from 151/100,000 in 1997 to 184/100,000 in 2017.

- "Previous literature has provided evidence on the lack of effectiveness of APM in patients with a degenerative knee or arthritis as the symptoms may be caused by the degeneration and not the meniscal tear." Please add reference.

A reference has been added.

- International consensus from specialist knee societies has highlighted the importance of patient factors which are important in the management of meniscal tears. Please add reference, and if you choose to add the reference from your own groups (17), change 'international' to 'national'.

We have changed to national.

- "For example, patients with a particular pattern of meniscal tear visible on MRI ('a meniscal target') and the corresponding pattern of symptoms or signs may benefit from APM." – there is hardly any robust evidence for this. Please delete this statement.

We have corrected this statement to match the treatment guidelines and emphasised this is based on guidelines.

- "Previous evidence supports the view that APM may not be beneficial in patients with meniscal tears and degeneration or arthritis within the knee joint (18)." This is incomplete. Several RCT have published their results and found no difference between APM and PT (or SHAM surgery) in patients with meniscal tears (Katz,Sihvonen,Yim,Kise,Herrlin,vdGraaf,Gauffin). None of the studies found clinically relevant differences between groups (with or without OA).

This statement has been corrected and moved to the discussion

- There is a large number of self-references (11 times to 6 studies). For example, "This could explain why the literature reports that up to 30% of patients cross over into the surgery group (4)." Why do refer to you own systematic review here? This does not seem logic.

We have referenced work carried out by the University of Oxford knee group, however, this is not from our own group. Only Professor Price is included in both groups. The systematic review was not carried out by ourselves. Another reason for including 6 references from Abram et al is because they have produced up to date evidence on the incidences and trends of arthroscopic meniscectomy in the UK. In addition, they were the lead authors for the BASK consensus statement. As this work is predominantly being carried out at the University of Warwick CTU we do not believe these are self-references. We should consider high-quality work that is directly relevant to the study, regardless of its origin. We should also note that we have heavily referenced other prominent research groups in this field as well.

- "The Knee (BASK) guidelines suggest that radiology investigations should be included in treatment." please add reference.

Reference added.

Methods:

- How will the authors determine the level of osteoarthritis? Only on MRI (which is known to give an overestimation and also is highly unfeasible due to its extensive nature, costs, heterogeneous outcome reporting)? or will there also be an X-ray?

Osteoarthritis will be determined based on MRI. These will be the clinical images which are routinely used in the UK and the basis of most surgeons normal decision making, and therefore directly applicable to practice. We will report the WOMS score for all included patients.

We are using standard of care MRIs which all patients will have as part of routine clinical care. Therefore there will be no additional costs. We believe this approach is feasible as patients in this age group with symptoms of a meniscal tear are routinely referred for a MRI scan. In secondary care, a common pathway is for patient to present to a specialist with an MRI performed. We do not plan on performing additional Xrays or MRIs for research purposes, we are only planning on using standard of care imaging. X-ray would give a poor assessment of structural disease and requiring sites to do x-rays would have both a cost and a radiation burden which is not necessary.

- Who will report the MRI? Several radiologists or only one or two that are trained according to the study protocol?

MRIs will be reported by a consultant radiologist (CEH), with extensive experience in the reporting and scoring of knee MRI scans for radiological and research purposes, and an orthopaedic surgical trainee (IA) who will be trained by a CEH to a high standard to report WOMS scores.

- "PROMs will be collected at recruitment, as a baseline for PROMS at 12 months (primary outcome). With follow up will be at three months, six months and 12 months." Please change sentence for better understanding.

This has been corrected.

- Can the authors explain why they chose for the KOOS instead of the IKDC as secondary outcome? If I interpret their own systematic review (and also the literature), I have come to conclude that the IKDC performs better on all domains compared to the KOOS. Most importantly, it is less than half of the length of the KOOS.

The reason for selecting KOOS as our secondary outcome measure is to allow comparisons to be made with the well-established KACS cohort. We take on the point regarding the length of KOOS vs IKDC, this was presented to our PPI group and the conclusions of the discussion was the KOOS score length was acceptable for the final questionnaire (12 months).

- Sample size calculation: why do the authors not account for crossovers in their sample size calculation?

Our principal aim is to describe factors which affect outcome in patients with a meniscal tear at the point of diagnosis (presentation to secondary care). As a result we did not include crossovers in the sample size calculation. Participants undergoing surgery is an outcome measure of interest as associations can be explored between baseline factors and need for surgery.

- Please elaborate on how the MIC will be determined. Will ROC curves be determined for an optimal cut-off point? This needs more attention. The difference in the mean WOMET score will represent the MCID. I am really not convinced that this is the most valid method. Please check the COSMIN guidelines, as an incorrect calculation of the MIC will be of huge impact on the interpretation of the study results.

We plan on using a longitudinal anchor based approach. The mean change in WOMET score will be calculated by subtracted the 12-month score from the baseline WOMET score. Participants will respond to an anchor question at 12 months which will contain 7 anchors. The mean change in WOMET scores will be calculated for each of the 7 anchors. We will then use ROC to determine the MCID with equal sensitivity and specificity to discriminate between the unchanged and the improved category (changed category is 5-7 and unchanged is 1-4). The area under the ROC curve represents the probability the score discriminates between the unchanged and the changed group. A probability of >0.7 was deemed acceptable.

Patient involvement:

- Well done! My compliments that you have been able to involve patients from the start.

Thank you

- "Eighteen patients with knee pain or osteoarthritis" Did these patients have a meniscal tear? This information is imperative.

Twelve patients had a meniscal tear. The others had osteoarthritis.

Data management:

- "Participants will be identified by ID number initials." Who is responsible for the ID, what will the ID comprise of (any retrievable information?) And who will have access to the ID key? Please elaborate on this.

This has been changed

Discussion section is missing. Please add this section and add relevant information, such as strengths/limitations, comparison with literature (there is one trial that is currently analyzing their

results (the STARR trial), which has the same target group; please cite this study. Please stress that this observational study can generate hypothesis for future studies. Causal inference is limited “It will also allow a large randomised study to be planned to determine the best treatment for young people who are diagnosed with meniscal tears.” Why do the authors not elaborate on this? It seems only reasonable to at least mention this as aim of the study, the discussion section seems appropriate for this.”

A discussion has been added, we have been clear this is hypothesis generating and have included explanations around many of the issues mentioned above.

VERSION 2 – REVIEW

REVIEWER	Jonas Bloch Thorlund University of Southern Denmark, Denmark
REVIEW RETURNED	03-Jun-2020

GENERAL COMMENTS	Thank you for clarifying my questions. Good luck with the study.
--

REVIEWER	Rudolf Poolman and Victor van de Graaf OLVG Amsterdam
REVIEW RETURNED	09-Jun-2020

GENERAL COMMENTS	Thank you for your extensive revision and elaboration on many of my comments. Being aware that this is an ongoing cohort, your answers have provided further clarification on any of my ambiguities. I have only one final comment with regard to the sample size. Although the authors clearly describe their aim (to describe factors which affect outcome in patients with a meniscal tear at the point of diagnosis), I am worried that the calculated sample size will limit the interpretation of their results. This is considered only as food for thought and clearly not a reason for a further revision. My congratulations on this fine work.
---